# Unsupervised Representation Learning of Medical Images for Downstream Segmentation

Mingrui Zhuang[1] and Hongkai Wang[1,2]

[1] School of Biomedical Engineering, Faculty of Medicine, Dalian University of Technology, Dalian 116024, China
[2] Liaoning Key Laboratory of Integrated Circuit and Biomedical Electronic System
`wang.hongkai@dlut.edu.cn`

**Abstract.** Automatic abdominal organ and tumor segmentation from CT scans can enhance clinical diagnosis and treatment planning, but manual annotation remains predominant due to limitations in current automated methods' robustness and accuracy. We propose a novel representation learning approach that trains a large-scale anatomical positional encoding network (LAPN) in an unsupervised manner, overcoming limited labeled data. LAPN encodes pixel-level anatomical localization information to guide downstream segmentation. We employed a U-Net network that takes in both the original image and positional encoding features to accomplish the specific segmentation task. Due to the large size of the LAPN segmentation pipeline, we use knowledge distillation to train a lightweight U-Net for efficient inference. The experiments demonstrate that LAPN can leverage unlabeled data to improve the performance of the segmentation network, particularly for organs with relatively fixed anatomical positions.

**Keywords:** Medical image segmentation· Representation learning · Incomplete supervision learning.

## 1 Introduction

Automatic multi-organ segmentation of abdominal computed tomography (CT) scans has the potential to enhance clinical workflows including diagnosis, prognostic analysis, and treatment planning for related diseases. However, manual annotation remains the predominant approach in current clinical practice [4], despite being tedious and prone to subjective variability amongst users. A key factor limiting clinical adoption of automated segmentation methods is their inferior robustness and accuracy compared to manual approaches. Moreover, most existing methods are focused on specific tumors rather than providing broadly generalizable solutions for comprehensive abdominal organ and tumor segmentation [10, 13]. Sufficient labeled training data, semi-supervised and unsupervised learning techniques, and model optimization will be critical to develop universally applicable automated segmentation models. Additionally, computational efficiency and minimal resource requirements are necessary to facilitate clinical

deployment, as exceedingly large, resource-intensive models are impractical for implementation.

In recent years, increased computing power have enabled significant progress in research on large-scale deep learning models, including large language models (LLMs) [12, 20] and large visual models [11]. Such large-scale models also hold great promise for medical image processing, as the vast quantity of medical images available can be leveraged to train the models. Some research applying large-scale models specialized for medical images has already shown promising performance. One approach involves extending general large-scale visual models to medical images. For example, the Segment Anything (SAM) model [11], trained on over 11 million diverse images, demonstrates strong generalization and interactivity on medical data. Fine-tuning or adapting SAM's architecture for medical image segmentation has achieved excellent results thus far [15]. Another approach is to train large models from scratch using medical imaging data. For example, STU-Net [8] trained a large-scale model with 14 million to 1.4 billion parameters on extensive medical image segmentation datasets, demonstrating that increasing model size leads to greater performance gains.

Moreover, training large-scale models for medical images faces the challenge of limited annotations. Unlike natural images, acquiring annotations for medical images is far more laborious, costly, and sometimes infeasible. This highlights the importance of unsupervised and semi-supervised training, which can mitigate the annotation burden. Through unsupervised training, models can leverage massive amounts of unlabeled medical image data. For example, models pretrained with unsupervised learning can initialize downstream networks [25], or unsupervised proxy tasks can facilitate training the target network. Additionally, a promising approach is to separately train an unsupervised feature extraction network [2], where the learned features are then utilized downstream. This decouples large model training from downstream objectives. Compared to unsupervised training, semi-supervised training can better tailor the model for the target task. When training large models, effectively leveraging previously accumulated image labels to improve performance remains an open problem. Many methods utilizing limited supervised data and abundant unlabeled data have been proposed, but most are designed for specific scenarios. Effectively incorporating semi-supervised learning to exploit existing labels during large model training remains an important open problem.

In this work, we propose a representation learning approach to train a foundational model that encodes anatomical positional information for input image pixels. This model can guide downstream segmentation networks and is trained in an unsupervised manner, enabling the use of large datasets to scale up the network size. The obtained anatomical positional encodings from the network, along with the input images, are fed into the downstream segmentation network during training. While the anatomical encoding network and segmentation network together can complete a full segmentation task, the combined network size is too large for practical deployment. To address this, we employ knowledge dis-

tillation to train a small U-Net segmentation network as the final deployment model for inference. Our method has the following main features:

– We train a large-scale anatomical positional encoding network (LAPN) using contrastive learning. Optimizations to the training pipeline and supervision paradigm allow utilizing larger datasets. The learned encodings can guide various downstream tasks like segmentation and registration.
– We trained a downstream segmentation network that integrates anatomical localization features generated by LDPN with the original CT images. Furthermore, we employed knowledge distillation to train a lightweight segmentation network, using LDPN and the downstream network as teacher models. This aimed to enhance inference efficiency while transferring anatomical localization knowledge.

## 2    Method

The workflow of our method is illustrated in Fig 1. We first train a large-scale anatomical positional encoding network in an unsupervised manner using contrastive learning. Inspired by self-supervised learning of pixel-wise anatomical embeddings [23], we supervise the model during training to output identical encodings for pixels at the same anatomical locations. This enables the network to learn to recognize anatomical positions. The downstream segmentation network is designed to take both input images and positional encodings as inputs simultaneously. Its training utilizes annotated examples for supervision, but since the encoding network is trained on the entire dataset, the segmentation network also benefits from unlabeled data. At this point, collaboration between the encoding and segmentation networks allows completing the full segmentation task, although the combined model size makes clinical deployment impractical. To reduce computational demands and improve inference speed, we employ knowledge distillation to train a lightweight U-Net model. The encoding and segmentation networks together serve as teacher networks, distilling their knowledge into the U-Net student network. This yields a significantly reduced U-Net size while maintaining performance, thereby accelerating inference.

### 2.1    Preprocessing

A simple preprocessing pipeline is adopted, which mainly includes the following steps:

– All images were resampled to a resolution of 1mm×1mm×1mm.
– The grayscale values of the image were limited to the range of [-500, 700] using a clamp operation, and then mapped to the range of [-1, 1].
– The preprocessed images are saved in an uncompressed format on a solid-state drive (SSD) for quick retrieval during training.

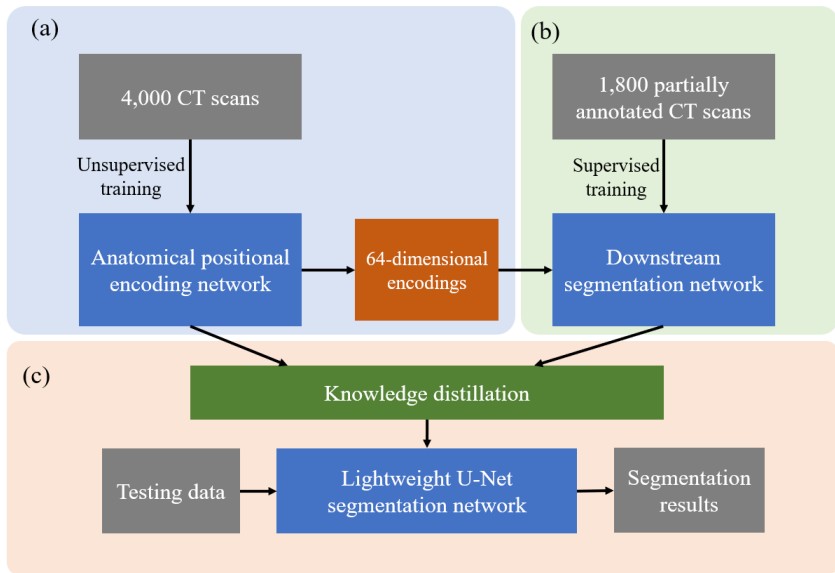

**Fig. 1.** Workflow of the proposed method. (a) An anatomical positional encoding network trained from a large amount of unlabeled data. The network is trained using unsupervised contrastive learning. The network's output consists of 64-dimensional anatomical position encodings for each pixel in the input image. (b) A downstream segmentation network that takes both CT images and anatomical position encodings as inputs. (c) A lightweight segmentation network trained through knowledge distillation for deployment purposes.

### 2.2 Proposed Method

**Large-scale anatomical positional encoding network (LAPN).** LAPN utilizes anatomical encoding tasks and momentum contrastive learning (MoCo) [5] in its architecture, as illustrated in Fig 2. In the first step, two overlapping image patches are randomly cropped from the input data. The arbitrary points in the overlapping region will be located at different positions in the two patches, as denoted by the red dots in Fig 2. The patches are then fed into a 5-level U-Net structured anatomical positional encoding network, yielding 64-dimensional feature maps. Each pixel's 64D feature represents the anatomical positional characteristics of that point. During training, contrastive learning supervises the network to produce similar feature distances between the same anatomical points in different patches, while keeping the distances between different points far apart.

Momentum contrastive learning is also adopted to accelerate training. Specifically, two identically structured encoding networks are maintained during training, e.g. network Q and network K. Network Q is trained normally, while network K's parameters are updated towards network Q using momentum updates. Once a patch pair is fed into network Q and network K, their outputs serve as positive sample pairs. The output of network Q and the previous output of network K are negative sample pairs. The loss is computed on these pairs, updating the networks accordingly. Network K's output is then recorded and used as negative samples for subsequent training.

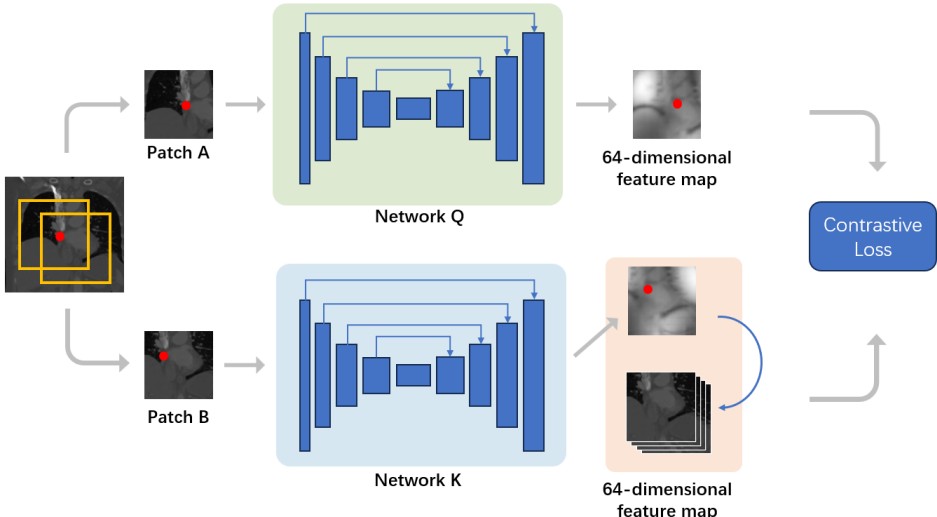

**Fig. 2.** The learning process of LAPN

**Downstream segmentation network.** We utilize a 5-level U-Net as the downstream network to leverage the anatomical positional encodings generated

by LAPN for segmentation. To enable simultaneous input of position encodings and original images, we modified the U-Net by adding another initial layers to the encoder. The features from the two initial layers are concatenated and fed into the subsequent U-Net architecture as a combined input. We adopt a composite loss function comprising Dice and cross-entropy losses, as established in prior work [14]. For partially labeled data, missing labels are simply ignored when calculating the loss. This means that if the network outputs missing labels in the background region of the label map, it will not be penalized.

**Strategies to improve inference speed and reduce resource consumption.** The large scale of LAPN and the downstream segmentation networks impedes deployment. To address this, we employ knowledge distillation to train a lightweight segmentation network. LAPN and the downstream network serve as teacher models, while the student adopts a 5-level U-Net architecture with no residual units per level. Specifically, the distillation temperature is gradually increased from 1 to 4 during training. The model is deployed using ONNX Runtime to further enhance inference speed.

## 3 Experiments

### 3.1 Dataset and evaluation measures

The FLARE 2023 challenge is an extension of the FLARE 2021-2022 [16] [17], aiming to aim to promote the development of foundation models in abdominal disease analysis. The segmentation targets cover 13 organs and various abdominal lesions. The training dataset is curated from more than 30 medical centers under the license permission, including TCIA [3], LiTS [1], MSD [22], KiTS [6,7], and AbdomenCT-1K [18]. The training set includes 4000 abdomen CT scans where 2200 CT scans with partial labels and 1800 CT scans without labels. The validation and testing sets include 100 and 400 CT scans, respectively, which cover various abdominal cancer types, such as liver cancer, kidney cancer, pancreas cancer, colon cancer, gastric cancer, and so on. The organ annotation process used ITK-SNAP [24], nnU-Net [9], and MedSAM [15].

The evaluation metrics encompass two accuracy measures—Dice Similarity Coefficient (DSC) and Normalized Surface Dice (NSD)—alongside two efficiency measures—running time and area under the GPU memory-time curve. These metrics collectively contribute to the ranking computation. Furthermore, the running time and GPU memory consumption are considered within tolerances of 15 seconds and 4 GB, respectively.

### 3.2 Implementation details

**Environment settings** The development environments and requirements are presented in Table 1.

**Table 1.** Development environments and requirements.

| System | Ubuntu 22.04.2 LTS |
|---|---|
| CPU | 2×Intel(R) Xeon(R) Platinum 8358P CPU @ 2.60GHz |
| RAM | 16×32GB; 3.2MT/s |
| GPU (number and type) | 8×NVIDIA A800 80G |
| CUDA version | 11.7 |
| Programming language | Python 3.10.10 |
| Deep learning framework | torch 2.0.0, torchvision 0.15.1 |
| Specific dependencies | monai 1.1.0 |

**Training protocols** To directly evaluate the proposed methods, we employed simple training protocols without data augmentation or additional processing of unlabeled and partially labeled data. During LAPN training, we randomly selected patch pairs with at least 1/8 overlap for contrastive learning. These patches also underwent random scaling up to 10% to further increase variability during contrastive learning, as detailed in Table 2. The training strategies for the downstream segmentation and knowledge distillation networks were relatively simpler, involving no data augmentation. Instead, random fixed-size patches were extracted for training, as detailed in Tables 3 and 4.

**Table 2.** Training protocols for the LAPN.

| Network initialization | Random |
|---|---|
| Batch size | 1 |
| Patch size | 192×192×192 |
| Total epochs | 500 |
| Optimizer | Adam |
| Initial learning rate (lr) | 1e-4 |
| Lr decay schedule | Cosine Annealing Lr |
| Training time | 73 hours with 8 GPUs |
| Loss function | InfoNCE loss [19] |
| Number of model parameters | 44.81 M[3] |
| Number of flops | 1535.12 G[4] |

## 4 Results and discussion

### 4.1 Quantitative results on validation set

Table 5 displays the Dice and NSD scores for organ and tumor segmentation attained by our proposed method on the validation set. All organs achieved Dice scores above 70, with liver, kidney, and spleen surpassing 90 due to being more

**Table 3.** Training protocols for the downstream segmentation networks.

| | |
|---|---|
| Network initialization | Random |
| Batch size | 1 |
| Patch size | 192×192×192 |
| Total epochs | 1000 |
| Optimizer | Adam |
| Initial learning rate (lr) | 1e-4 |
| Lr decay schedule | Cosine Annealing Lr |
| Training time | 44 hours with 1 GPU |
| Loss function | DiceCELoss |
| Number of model parameters | 10.94 M[5] |
| Number of flops | 285.00 G[6] |

**Table 4.** Training protocols for the knowledge distillation networks.

| | |
|---|---|
| Network initialization | Random |
| Batch size | 1 |
| Patch size | 192×192×192 |
| Total epochs | 1000 |
| Optimizer | Adam |
| Initial learning rate (lr) | 1e-4 |
| Lr decay schedule | Cosine Annealing Lr |
| Training time | 24 hours with 1 GPU |
| Loss function | DiceCELoss |
| Number of model parameters | 7.94 M[7] |
| Number of flops | 6.37 G[8] |

readily segmented. However, tumor segmentation results were generally inferior. The weaker correlation between tumors and anatomical locations may limit the utility of LAPN for assisting the segmentation network. Further improvements to the anatomical encoding could enhance tumor localization and boost segmentation performance.

**Table 5.** Quantitative evaluation results.

| Target | Public Validation | | Online Validation | | Testing | |
|---|---|---|---|---|---|---|
| | DSC(%) | NSD(%) | DSC(%) | NSD(%) | DSC(%) | NSD (%) |
| Liver | 97.75 ± 0.66 | 87.94 ± 4.02 | 98.03 | 97.41 | | |
| Right Kidney | 94.33 ± 6.87 | 89.02 ± 11.68 | 93.59 | 93.94 | | |
| Spleen | 96.88 ± 2.68 | 93.61 ± 6.63 | 96.89 | 97.98 | | |
| Pancreas | 83.17 ± 6.46 | 68.18 ± 9.41 | 80.38 | 93.67 | | |
| Aorta | 87.06 ± 8.42 | 75.10 ± 12.85 | 87.92 | 82.91 | | |
| Inferior vena cava | 89.12 ± 3.71 | 80.36 ± 6.74 | 89.74 | 91.53 | | |
| Right adrenal gland | 76.06 ± 4.71 | 76.61 ± 16.88 | 73.38 | 86.34 | | |
| Left adrenal gland | 71.00 ± 8.33 | 73.49 ± 10.40 | 70.11 | 83.20 | | |
| Gallbladder | 74.27 ± 24.04 | 54.58 ± 26.39 | 71.47 | 68.06 | | |
| Esophagus | 74.69 ± 14.80 | 67.40 ± 13.82 | 76.61 | 89.50 | | |
| Stomach | 88.38 ± 9.69 | 63.50 ± 14.17 | 88.51 | 90.25 | | |
| Duodenum | 74.88 ± 10.78 | 59.44 ± 10.82 | 74.54 | 91.83 | | |
| Left kidney | 91.97 ± 14.28 | 86.87 ± 17.51 | 92.68 | 92.78 | | |
| Tumor | 30.41 ± 33.19 | 14.99 ± 18.78 | 25.35 | 16.67 | | |
| Average | 80.71 ± 10.62 | 70.79 ± 12.86 | 84.09 | 89.23 | | |

## 4.2   Qualitative results on validation set

Fig 3 presents qualitative results on the validation set. For Case #FLARETs_0003, the network with LAPN anatomical landmark features as input achieves improved pancreas segmentation compared to the network without these features. However, for Case #FLARETs_0009, our method demonstrates poorer stomach segmentation. This inferior performance may stem from the air-filled stomach regions confounding the anatomical landmark features with low pixel intensities.

## 4.3   Segmentation efficiency results on validation set

To enable fast inference and reduced resource requirements for practical deployment, we leverage knowledge distillation to train a lightweight U-Net model. This compressed network is well-suited for real-world application. The model's performance on the test dataset is summarized in Table 6. With the exception of '0029', the inference time for all other instances is within one minute. Knowledge distillation effectively minimizes model size while retaining original segmentation capability. The lightweight architecture substantially reduces inference time, satisfying real-time deployment needs.

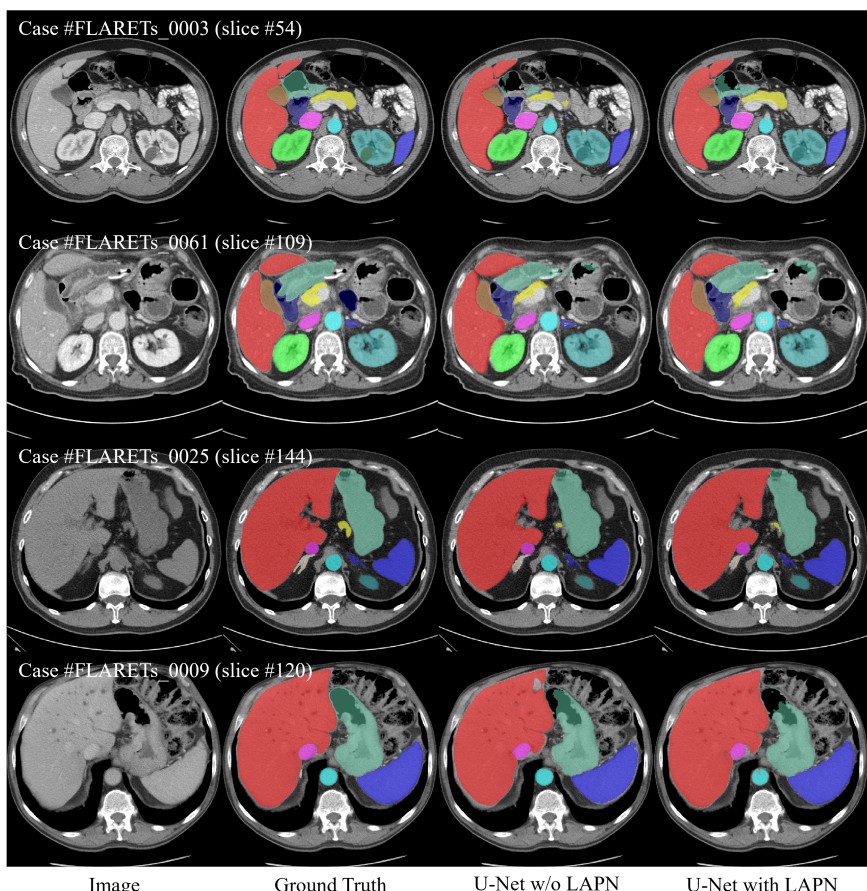

Case #FLARETs_0003 (slice #54)

Case #FLARETs_0061 (slice #109)

Case #FLARETs_0025 (slice #144)

Case #FLARETs_0009 (slice #120)

| Image | Ground Truth | U-Net w/o LAPN | U-Net with LAPN |

**Fig. 3.** Qualitative results on validation set.

**Table 6.** Quantitative evaluation of segmentation efficiency in terms of the running them and GPU memory consumption. Total GPU denotes the area under GPU Memory-Time curve. Evaluation GPU platform: NVIDIA QUADRO RTX5000 (16G).

| Case ID | Image Size | Running Time (s) | Max GPU (MB) | Total GPU (MB) |
| --- | --- | --- | --- | --- |
| 0001 | (512, 512, 55) | 33.45 | 6402 | 61241 |
| 0051 | (512, 512, 100) | 41.13 | 6402 | 100034 |
| 0017 | (512, 512, 150) | 45.06 | 3310 | 109114 |
| 0019 | (512, 512, 215) | 38.97 | 3310 | 94117 |
| 0099 | (512, 512, 334) | 48.98 | 6464 | 122405 |
| 0063 | (512, 512, 448) | 59.90 | 3308 | 147304 |
| 0048 | (512, 512, 499) | 59.97 | 6472 | 150748 |
| 0029 | (512, 512, 554) | 70.62 | 6400 | 176032 |

### 4.4    Results on final testing set

This is a placeholder. We will send you the testing results during MICCAI (2023.10.8).

### 4.5    Limitation and future work

Results show LAPN provides modest assistance for downstream segmentation, largely due to insufficient accuracy in anatomical position encoding. This limitation constrains LAPN's applicability. Future work should focus on improving LAPN performance through enhancements in network architecture, training methodology, and other avenues. Realizing LAPN's full potential would significantly increase its value for downstream medical image analysis tasks.

## 5    Conclusion

This study presents a novel representation learning approach for training foundational models in medical image analysis. Our method enables unsupervised training of such models using abundant unlabeled data. We employ an anatomical position encoding task with momentum contrast learning to efficiently train the network. The extracted anatomical localization features are input to downstream segmentation networks, improving segmentation accuracy. Furthermore, knowledge distillation trains a compact final model, balancing inference speed and resource efficiency. Experiments demonstrate this approach leverages unsupervised data to enhance abdominal organ segmentation performance.

**Acknowledgements** The authors of this paper declare that the segmentation method they implemented for participation in the FLARE 2023 challenge has not used any pre-trained models nor additional datasets other than those provided by the organizers. The proposed solution is fully automatic without any manual intervention. We thank all the data owners for making the CT scans publicly available and CodaLab [21] for hosting the challenge platform.

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
