# OpenReview forum: "Unsupervised Representation Learning of Medical Images for Downstream Segmentation"
_MICCAI.org/2023/FLARE — Submitted to FLARE 2023_

### Official Review · Reviewer_MNQn · 2023-10-03
**Review for“Unsupervised Representation Learning of Medical Images for Downstream Segmentation”**

**Rating:** 6
**Confidence:** 4

**Review:**

Pros:
This paper effectively trains the models by using a large amount of unlabeled data through unsupervised learning. It employs an anatomical position encoding task with momentum contrastive learning to train the network efficiently. Extracted anatomical location features are input into downstream segmentation networks, improving segmentation accuracy. Furthermore, a compact final model is trained with knowledge distillation, balancing inference speed and resource efficiency. This approach achieves an average DSC of 80.71% in multi-organ abdominal segmentation and an average DSC of 25.35% in pan-tumor segmentation.
Cons:
1. ORCID for authors has not been annotated as required.
2. The quantitative results of NSD and area under the GPU memory time curve are not presented in abstract.
3.Fig. 2 does not use the Times New Roman font, and it would be beneficial to include an explanation of the LAPN process.
4.In Table 1, GPU (number and type) does not specify specific sequence numbers.
5.Chapter four needs to address the use of unlabeled data and perform ablation experiments.
6.There is no Checklist Table in the paper.

---

### Official Review · Reviewer_RtSX · 2023-10-04
**Good writing but lack of some details**

**Rating:** 6
**Confidence:** 4

**Review:**

Pros: This paper uses unsupervised learning with labeled and unlabeled data to train the network, and employs the U-Net network architecture, to complete the specific segmentation tasks by receiving original images and position-encoded features. Knowledge distillation is applied to train a lightweight U-Net for efficient inference. An average DSC of 80.71% in multi-organ abdominal segmentation and an average DSC of 25.35% in pan-tumor segmentation are achieved.

Cons:
1.ORCID numbers for multiple authors are not annotated.
2.The abstract does not reflect required metrics such as DSC.
3. Fig.2 do not use the Times New Roman font.
4.Suppressed title at the top of the page have not been defined.
5.In Table 1, the "GPU (number and type)" field does not indicate the sequence number of the GPUs used.
6.The use of unlabeled data needs explanation and there is no content related to ablation experiments.
7.There is no Checklist Table in the paper.

---

### Official Review · Reviewer_SmtL · 2023-10-04
**Potential for Enhanced Performance through Hyperparameter Optimization in Anatomical Segmentation Approach**

**Rating:** 7
**Confidence:** 4

**Review:**

While the paper's unique approach to abdominal organ and tumor segmentation, considering anatomical characteristics, was impressive compared to other papers, the performance was somewhat lacking. However, I anticipate that optimizing the hyperparameters could lead to improved results.

---

### Official Review · Reviewer_WojU · 2023-10-19
**More detail would be helpful**

**Rating:** 6
**Confidence:** 4

**Review:**

Pros: The author draws inspiration from MoCo's unsupervised contrastive learning method and innovatively proposes a method that combines unsupervised features with partially labeled information to train a complete segmentation model. Additionally, they use knowledge distillation to obtain a lightweight deployment model. The problem description is clear, and the solution is novel and effective.

Cons: 1.Providing more details on the implementation process and showcasing additional results would further convince readers of the method's effectiveness.2.It is recommended to provide a more specific introduction to the methods used for knowledge distillation.3.Conducting ablation experiments and comparative experiments with different models would make the algorithm's effectiveness more pronounced.

---

### Official Review · Reviewer_LpvD · 2023-10-20
**More detail would be helpful**

**Rating:** 6
**Confidence:** 5

**Review:**

This paper presents an unsupervised representation learning method for downstream medical image segmentation, inspired by MOCO. Subsequently, a lightweight segmentation network is proposed based on knowledge distillation. While the overall approach is feasible, it is regrettable that the segmentation results, especially for tumor segmentation, appear to be somewhat subpar. Further consideration should be given to the characteristics of pan-cancer segmentation tasks. Additionally, the paper lacks detail in the methodology, particularly concerning self-supervision and knowledge distillation, which should be further elaborated and clarified.

---

### Public Comment · ~PENGJU_LYU1 · 2023-11-26
**add test results**

fill in the table 5 with test results

---

### Decision · Program_Chairs · 2023-10-25

**Decision:**

Reject

**Comment:**

The authors didn't make responses to the valuable review comments.